# Role of the Gut–Liver Axis in the Pathobiology of Cholangiopathies: Basic and Clinical Evidence

**DOI:** 10.3390/ijms24076660

**Published:** 2023-04-03

**Authors:** Maria Consiglia Bragazzi, Rosanna Venere, Anthony Vignone, Domenico Alvaro, Vincenzo Cardinale

**Affiliations:** 1Department of Medical-Surgical Sciences and Biotechnology, Sapienza University of Rome Polo Pontino, 04100 Roma, Italy; 2Department of Translational and Precision Medicine, Sapienza University of Rome, 04100 Roma, Italy

**Keywords:** Gut liver Axis, microbiota, choalngiopathies

## Abstract

The “Gut–Liver Axis” refers to the physiological bidirectional interplay between the gut and its microbiota and the liver which, in health, occurs thanks to a condition of immune tolerance. In recent years, several studies have shown that, in case of a change in gut bacterial homeostasis or impairment of intestinal barrier functions, cholangiocytes, which are the epithelial cells lining the bile ducts, activate innate immune responses against gut-derived microorganisms or bacterial products that reach the liver via enterohepatic circulation. Intestinal dysbiosis or impaired intestinal barrier functions cause cholangiocytes to be exposed to an increasing amount of microorganisms that can reactivate inflammatory responses, thus inducing the onset of liver fibrosis. The present review focuses on the role of the gut–liver axis in the pathogenesis of cholangiopathies.

## 1. The “Gut–Liver Axis”

The concept of the “gut–liver axis”, was introduced by Marshall in 1998, and it refers to the close anatomical and functional relationship between the gut and the liver. The interaction between these organs is bidirectional and is characterized by the transfer, to the liver, of molecules associated with the intestinal microbiota, starting from the intestine, thanks to the vascular pathway of the portal vein, and, at the same time, by the hepatic feedback pathway of bile and antibody secretion in the intestine [1]. In health, there is a mechanism defined as “immune intolerance,” thanks to which part of the intestinal mycorbic antigens enter the portal blood circulation and are recognized by the immune cells of the liver without eliciting an immunological response [2].

In particular, in the gut, the mucosal barrier separates the host immunity and the intestinal microbiota in order to avoid an unfavorable interaction between the two. The luminal side of the intestine is lined with epithelial cells which, promote the absorption of water and nutrients, and also play an important role in generating a physical and chemical barrier to protect the mucosa from commensal and pathogenic microorganisms [3]. Regenerating islet-derived protein (Reg3γ) and cyclic adenosine monophosphate (AMP) secreted by Paneth cells constitute the chemical barrier and mainly contribute to the separation between intestinal bacteria and intestinal epithelial cells in the small intestine [3]. Instead, in the large intestine hosts the inner mucus layer, composed of polymerized mucin Mucin 2, oligomeric mucus gel-forming (MUC2), which acts as a physical barrier and separates intestinal bacteria from intestinal epithelial cells [3]. The stability of this equilibrium is essential in maintaining intestinal homeostasis [3,4,5]. The gut microbiota is involved in promoting the digestive process, nutrient absorption, and the production of small chain fatty acids (SCFA). It is also a primary energy source for intestinal epithelial cells (IEC), promoting the stimulation of immune responses through the release of ligands and providing protection against enteropathogens through the production of antimicrobial peptides (AMPs), which serve as a protective barrier against pathogens from competing for space and food [6,7]. 

The composition of the gut microbiome is influenced by several factors including diet, age, host genotype, disease state, and antibiotic exposure [7]. The composition of the intestinal microbiota is different for each individual, and there is no optimal composition, even if a healthy host–microorganism balance must be respected. This condition is defined by the term *eubiosis*, and allows metabolic and immune functions to be performed optimally.

In conditions where there is an imbalance in the gut microbial community, in terms of qualitative and quantitative changes, metabolic activity, and topographic distribution, we speak of *dysbiosis* of the gut microbiota [8]. Dysbiosis can disrupt the integrity of the mucosal barrier [9,10]. Indeed, pathological conditions (e.g., a high-fat/high-fructose diet, intestinal inflammation, systemic diseases) promote dysbiosis and increase the permeability of the cellular pathway (tight junction disruption), causing inadequate nutrient absorption and the inability to prevent the translocation of luminal bacteria and their products (also called pathogen-associated molecular patterns or PAMPs), and the reabsorption of damage-associated molecular patterns (DAMPs) [11,12,13,14]. Typically, in conditions of dysbiosis, the hyperactivity of the innate immune response leads to an overexpression of M1/M2 type macrophages which, once activated, increase pro-inflammatory events. Moreover, regulatory T cells (Treg) regulate the adaptive immune response by maintaining tolerance to self-antigens and inducing a suppression of excessive activation of immune responses. Insufficient Treg expression can lead to increased levels of Th1 and Th17, facilitating chronic inflammatory responses. In turn, the gut barrier is heavily impaired (vicious circle), leading to systemic effects due to the translocation of bacteria and associated PAMPs or DAMPs [11,12,13,14]. 

In case of changes in intestinal bacterial homeostasis or in the presence of a disease state resulting in impaired intestinal barrier function, an altered composition of gut-derived products reaches the liver via the portal vein, potentially inducing liver inflammation which, in turn, generates subsequent complex responses in liver cells including cholangiocytes [15]. In the liver, microbes and pathogens drive inflammation by acting on receptors on liver sinusoidal endothelial cells, Kupffer cells, hepatic stellate cells, and by activating the NLRP3 inflammasome [16].

## 2. Biliary Barrier

Bile acids, oxysterols, antigens, endotoxins, and xenobiotics are bile constituents and the major determinants of this harmful bile microenvironment. It is known that the biliary system contrasts this environment with several biochemical systems. Among them, an important role is also exercised by alkaline phosphatase, which has an important function in endotoxin/xenobiotic detoxification. Recent evidence by our group has highlighted the visceral tissue organization of the bile ducts. We have previously discovered that biliary tree stem/progenitor cells (BTSCs) are located at the base of the peribiliary glands (PBGs) of large intrahepatic and extraheatic bile ducts, and that they differentiate into mature cholangiocytes and globet cells during migration from the bottom to the top of the glands. In summary, in the biliary tree, the intestinal model characterized by the proliferation of multipotent stem cells within the crypts of Lieberkuhn, by cell migration and by differentiation along the axis of the villi of the crypt, are fundamental for the development and maintenance of the architecture intestinal [17,18,19]. Evidence demonstrates that, like the gut, the biliary system also has a mechanical, chemical, and immunological barrier which ensures an immunological tolerance towards the commensals [14]. Indeed, the biliary epithelium also shows a wide range of innate immune receptors, such as Toll-like-receptor (TLR) 1 to TLR6 and TLR9. Antimicrobial peptides including human β-defensin-1 and -2 are widely expressed in the intrahepatic biliary tree. Tissue macrophages and liver Kupffer cells, activated by proinflammatory cytokines, induce microbial killing and antigen presentation to T cells and plasma cells in the biliary system. IgA are secreted in the bile by the biliary epithelium [14]. Moreover, specific microorganisms possess tolerance mechanisms in order to resist bile action. [4]. In the past, bile fluid was previously thought to be sterile. Several studies have documented the presence of bacteria even in the bile ducts in homeostatic conditions [11,12,13]. The existence of a biliary microbiome in healthy people was first documented in a study conducted by Molinero et al., through the collection of bile samples in patients undergoing gallbladder resections [12]. Results in experimental models on biliary injury demonstrate the crucial role played by the commensal microbiota and its metabolites in protecting against biliary injury and suggest avenues for future biomarker studies and therapeutic interventions in PSC [20].

## 3. Role of Bile Acids in Gut–Liver Interactions

Bile salts constitute a powerful tool in gut–liver bidirectional interactions and mediate systemic effects throughout their cognate cellular receptor, farnesoid X receptor (FXR), and tumor growth rate 5 (TGR5) [21,22].

FXR regulates the synthesis and transport of bile acids. It is a member of the nuclear metabolic receptor superfamily and, among many functions, by interacting with bile acids (BA), it regulates their synthesis and enterohepatic circulation [21,22]. Recent studies have shown that bile acids, after their transformation into secondary bile acids, signal intestinal epithelium primarily via the FXR. Primary BA from the gut lumen are transported into intestinal epithelial cells where they can bind to FXR, through which they promote the transcription of fibroblast growth factor 19 (FGF-19) [15]. 

TGR-5, also known as G protein-coupled bile acid receptor 1, is expressed in the liver, and in other organs, such as the intestine [23,24]. It is activated by bile acids, including cholic, chenodeoxycholic, deoxycholic, and lithocholic acid [25]. Similarly to TGR5, even FXR plays an important role in the enterohepatic circulation of cholic acid [26,27]. In turn, it has been documented that the microbiota can modulate signaling trough both FXR and TGR-5 via the modification of bile acids [28].

The liver communicates with the gut through the biliary system (the biliary tract connects the liver with the duodenum), and the systemic circulation by releasing bile acids (BA) and systemic inflammatory mediators like cytokines [29]. BAs are molecules synthesized in the liver from cholesterol and then released and reabsorbed in the gut by the microbiota [28]. The amount of BA produced depends on an active feedback loop from the gut to the liver, which is called the enterohepatic circulation [30]. Before being excreted, primary BAs are conjugated with the amino acid glycine and, to a lesser extent, with taurine in humans and subsequently released in the bile. About 90% to 95% of BAs are absorbed at the distal ileum and subsequently transported to the liver, where they are recycled after entering the portal circulation. The remaining parts of the BAs are degraded and biotransformed by microorganisms mainly at the level of intestinal tract, and some of them are excreted by feces [28]. The transformation from primary to secondary BAs (deconjugation and dihydroxylation) is facilitated by bile salt hydrolases (BSH) and 7α- dehydroxylase expressed by microbes of the gut microbiome including all major phyla (BSH) and the genera *Bacteroides*, *Clostridium*, *Eubacterium*, *Lactobacillus*, and *Escherichia* [15,28]. At the same time, BAs have a key role in shaping the microbiota. This indicates that there is a two-way interaction between bile acids and gut microbiota, as the microbiota affects the metabolism of bile acids [31] and bile acids affect microbiota composition.

## 4. The Gut–Liver Axis in Cholangiopathies

Cholangiopathies are a group of progressive diseases that affect cholangiocytes, the epithelial cells lining the small bile ducts (interlobular bile ducts; see Primary biliary cholangitis (PBC), Drug-induced liver injury (DILI), small bile duct intrahepatic cholangiocarcinoma (iCCA)) and the large bile ducts (septal, segmental, extrahepatic; see primary sclerosing cholangitis (PSC), IgG4-related cholangitis, large bile duct iCCA, perihilar cholangiocarcinoma (pCCA), distal cholangiocarcinoma (dCCA). These pathologies are generally chronic, and the pathogenetic processes affecting cholangiocytes in these cases are not yet fully known. Depending on their nature, these diseases are further subdivided into genetic; autoimmune, which includes PBC, PSC, and IgG4-related cholangitis; malignant, such as cholangiocarcinoma (CCA); or combined hepato-cholangiocarcinoma (CHC), and other categories [32,33]. Ascending cholestasis is a pathological consequence of cholangiopathies in which, although bile is produced by hepatocytes, the bile flow through the intestine is impaired, resulting in the intrahepatic accumulation of bile acids and a consequent state of inflammation in the liver and NF-kB-mediated production of pro-inflammatory cytokines [34,35,36,37]. Elevated levels of hydrophobic bile acids damage the bile duct epithelium and increase luminal pressure until the bile duct ruptures, resulting in hepatocytes being exposed to high concentrations of bile acids and inflammatory infiltration, and consequently, death of the hepatocytes [38]. 

In health, cholangiocytes are quiescent and participate to the final bile volume and composition. When normal gut bacterial homeostasis is disrupted, or in a case wherein the intestinal barrier function is defective, an altered composition of gut-derived products reaches the liver via portal vein where, potentially, they could induce or exacerbate hepatic inflammation, eliciting complex responses in hepatic cells including cholangiocytes that became reactive undergoing proliferation, senescence, and apoptosis. Immune and mesenchymal cell chemotaxis is also activated to repair damaged tissue and remodel the biliary tree [32,33]. In recent years, several studies on animal models have been conducted regarding the role played by the intestinal microbiota on chronic biliary inflammation [34,35,39,40]. As well as this, many studies have shown that bile duct ligation causes bacterial overgrowth with increased bacterial translocation [36,37,41]. In fact, cholangiopathies, which alter the composition or the normal flow of the bile, may interfere with all of the processes involving the gut and the liver, ultimately causing dysbiosis and an increased or qualitatively altered PAMP/DAMP delivery to the liver via the portal circulation. The chemical composition of the bile and physical properties are modulated in pathological conditions affecting the intestine, the liver, the cholangiocyte, and even in systemic metabolic conditions, obesity, or systemic inflammation [36,37,41]. Bile products (e.g., oxysterols) affect per se the biliary barrier and DAMPs and PAMPs from bile can recirculate to the liver via the peribiliary plexus, the anatomical vasculature framework underlining the biliary–liver axis [36,37,41]. 

## 5. PBC

PBC is a chronic autoimmune cholestatic liver disease characterized by cholestasis, serologic reactivity to antimitochondrial antibodies (AMA) or a specific antinuclear antibody (gp210 and sp100 antibody) reactivity, and it is histologically characterized by a chronic non-suppurative inflammation of the interlobular bile ducts (parenchymal ducts). In recent years, numerous studies have focused their attention on potential links between the gut microbiome and PBC. Thanks to this research, a different composition of the microbiota was observed in patients affected by this pathology, both in the gut and in the bile (see Table 1 and Table 2, respectively). 

In PBC, the alteration of the gut microbiome may be involved in the onset, progression, and prognosis. 

At pathobiology level, it has been proposed that AMAs, which bind to the mitochondrial E2 subunit of the pyruvate dehydrogenase complex (PDC-E2), may cross-react with bacterial proteins of Escherichia coli, Lactobacillus delbrueckii, and Novosphingobium aromaticivorans [42]. Indeed, the infection with Novosphingobium aromaticivorans of genetically susceptible mouse strains induces anti-PDC E2 responses and hepatic lesions similar to the typical lesions of human PBC. In the study Kitahata et al. [43] conducted in 34 patients with PBC and 21 healthy controls, the potential impact of the small intestinal mucosa-associated microbiota (MAM) in the pathogenesis of PBC was investigated. They performed 16S ribosomal RNA gene sequencing of MAM samples, obtained from the mucosa of the terminal ileum, and examined the relationship between the abundance of ileal MAM and chronic nonsuppurative destructive cholangitis. They concluded that dysbiosis of ileal MAM in PBC patients is characterized by an overgrowth of Sphingomonadaceae and Pseudomonas. Moreover, the abundance of Sphingomonadaceae is associated with chronic nonsuppurative destructive cholangitis in PBC, with a possible impact in the development of this disease. 

Another demonstration regarding the alteration of the gut microbiota and bacterial translocation associated with immune pathology was given by Hong-Di et al. in a murine model of PBC, dn Transforming growth factor-beta receptor II (dnTGFβRII) mice [44]. The authors demonstrated that the administration of antibiotics and the consequent alteration of microbiota, significantly alleviates T-cell-mediated infiltration and bile duct damage. Instead, the Toll-like receptor 2 (TLR2)-deficient dnTGFβRII mice showed exacerbation of autoimmune cholangitis when their epithelial barrier integrity was disrupted for a downregulated expression of tight junction-associated protein ZO-1, leading to increased gut permeability and bacterial translocation. Regarding the pathogenesis of PBC, Liwinski T et al. [45], in their recent review, underline that several large-scale, case–control studies indicate a significantly higher prevalence of recurrent urinary tract infections in patients with PBC, mainly caused by *Escherichia coli* that seems to be involved in the production of the disease-specific AMA (Table 1). Another candidate involved in the pathogenesis of PBC, according to Selmi et al., is the ubiquitous bacterium *Novosphingobium aromaticivoran*, which causes a potential break in tolerance to the self E2 component of the mitochondrial pyruvate dehydrogenase complex (PDC-E2) through two independent mechanisms [46]. Lv et al. confirmed the presence of an altered microbiota in PBC patients. This depleted some potentially beneficial bacteria, such as *Ruminococcus bromii*, and enriched potentially entailing pathogens such as phylum *Proteobacteria*, *family Enterobacteriaceae*, and the *genera Veillonella*, *Streptococcus*, and *Klebsiella* [47]. Additionally, in a study conducted by Tang et al., a large Chinese cohort of treatment-naïve PBC patients demonstrated a significant reduction of within-individual microbial diversity and an increase of potential pathogens, such as *Klebsiella*, *Haemophilus*, *Streptococcus*, and *Veillonella* in ursodeoxycholic acid (UDCA)-naïve PBC patients compared to controls with a restoration of balance after 6 months of treatment with UDCA [48]. 

Furukawa et al. focused their attention on a possible correlation between the response to UDCA, first-line therapy of PBC, and gut microbiome composition. They demonstrated that, in PBC patients treated with UDCA, persistence of gut dysbiosis could affect their clinical prognosis. When compared with healthy subjects, these patients had a decreased abundance of the order *Clostridiales* and increased abundance of *Lactobacillales*. The UDCA non-responder group had a significantly lower population of the genus *Faecalibacterium*, known as butyrate-producing beneficial bacteria, and this might predict the long-term prognosis of patients with PBC [49]. Another interesting aspect is represented by the effect of the bile acid sequestrant on icteric PBC subjects, as Cholestyramine, used to treat cholestatic pruritus. [50]. In their study, Bo Li et al. demonstrate how the administration of cholestyramine caused beneficial responses which were closely related with compositional and functional alterations in gut commensal. In fact, gut microbial co-abundance networks showed distinct taxa interactions between subjects with superior remission (SR) and those with inferior remission (IR) at baseline. After treatment, compositional shifts of the microbiome in the SR group are characterized by the enrichment of two Lachnospiraceae species, typically producing short-chain fatty acids (SCFAs), as confirmed by metabolome analysis [50]. 

Second-line therapy of PBC is currently represented by obeticholic acid (OCA), a first-order agonist that selectively binds to FXR, involved in the modulation of hepatic inflammation, fibrosis, metabolic pathways, and regeneration [51]. OCA acts directly and indirectly to suppress bile acid production in the liver and increase bile flow, with a consequent reduced exposure to toxic levels of bile acids. In mice with Bile Duct Ligation, FXR raised the expression of genes involved in enteroprotection and impeded bacterial overgrowth and mucosal injury in ileum. In addition, FXR activation via OCA represses chemically-induced intestinal inflammation in mice, suggesting the possible role of FXR in inflammatory bowel disease [52]. 

As far as the biliary microbiota is concerned, in a study, Hiramatsu et al. analyzed gallbladder bile samples from patients with PBC, PSC, hepatitis virus C-related liver cirrhosis choledocholithiasis, and normal adult gallbladders. They noted that, in patients with PBC, 75% (~0.0001) of 100 clones were identified as so-called Gram-positive cocci, while these cocci were positive in only 5% of cholecystolithiasis. Hence, this supports the hypothesis that Gram-positive bacteria may be involved in the etiopathogenesis of PBC, triggering bile duct inflammation, and/or antigen presentation [53] (Table 2).

**Table 1 ijms-24-06660-t001:** Changes in gut microbiota in cholangiopathies.

Primary Biliary Cholangitis	**Overrepresented**	**Decreased Abundance**
		*Acidobacteria* [47]
	*γProteobacteria* [47]	
	*Veillonella* [45,47,48] *Lactobacillales* [49]	*Clostidiales* [49]
	*Sphingomonadaceae* [43,47] *Pseudomonadaceae* [43] *Metylobacteriaceae* [43] *Moraxellaceae* [47,48] *Enterobacteriaceae* [47,48] *Neisseriaceae* [47]	
	*Haemophilus* [45] *Sterptococcus* [45,47] *Klebsiella* [45,47,48] *Actinobacillus* [47]	*Faecalimbacterium* [48] *Sutterella* [48] *Oscillospira* [48]
	*Anaeroglobus germinatus* [47] *Eterobacter asburiae* [47] *Hemophilus parainfluenzae* [47] *Megasphera micronuciformis* [47] *Paraprevotella Clara* [47] *Pleuropneumoniae* [47]	*Lachnobacterium* sp. [47]*. Bacteroides eggerthii* [47,48] *Ruminococcus bromii* [45,47]
Primary Sclerosing Cholangitis	*Veillonella* [54,55]	
	*Clostridium* [54] *Escherichia* [54] *Streptococcus* [45,55] *Enterococcus* [45,55]	*Eubacterium* spp. [54]
	*Clostridium cluster XIVa* [55] *B. producta* [55]	*Ruminococcus Obeum* [54] *Bacterioides thetaiotaomicron* [55] *Faecalibacterium prausnitzii* [55]
CCA	*Bacterioidetes* [56,57]	*Firmicutes* [56]
	*Veillonella* [56]	
	*Muribaculaceae* [57]	
	*Streptococcus* [56] *Klebsiella* [56] *Muribaculum* [57] *Alistipes* [57]	

Table 1**.** Taxonomy of gut microbiota in cholangiopaties. Legend: 
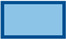
 PHYLUM 
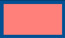
 ORDER 
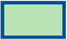
 GENUS 
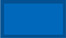
 CLASS 
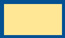
 FAMILY 

 SPECIES.

**Table 2 ijms-24-06660-t002:** Changes in biliary microbiota in cholestatic liver diseases.

Primary Biliary Cholangitis	**Overrepresented**	**Decreased Abundance**
	*Corynebacterium otitidis* [53]	
	*Staphylococcus aureus* [53] *Enterococcus faecium* [53] *Streptococcus pneumoniae* or other *streptococci* [53] *Lactohacillus plantarum* [53] *Helicobacter pylori* [53] *Propionibacterium acnes* [53] *Lactobacillus gasseri* [53] *Agrobacterium tumefaciens* [53] *Flavobacterium breve* [53] *Clostridium sordellii* [53] *Micrococcus luteus* [53]	
Primary Sclerosing Cholangitis	*Enterococci* [58] *Candida* [59]	
	*Enterococcus Faecalis* [58,60]	
pCCA/dCCA	*Bacteroidetes* [61] *Acidobacteria* [62] *Planctomycetes* [62]	*Firmicutes* [61]
	*Methylophilaceae* [62]	
	*Fusobacterium* [62] *Actinomyces* [62] *Novosphingobium* [62] *Enterococcus* [61] *Streptococcus* [61] *Klebsiella* [61] *Pyramidobacter* [61] *Geobacillus* [61] *Meiothermus* [61] *Anoxybacillus* [61]	*Nesterenkonia* [62] *Mesorhizobium* [62] *Rothia* [62]
	*Helicobacter Pylori* [62,63] *Prevotella* [62] *Helicobacter Bilis* [64]	

Table 2**.** Taxonomy of biliary microbiota in cholangiopaties. Legend: 
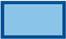
 PHYLUM 
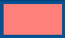
 ORDER 
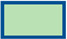
 GENUS 
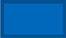
 CLASS 
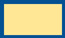
 FAMILY 

 SPECIES.

## 6. PSC

PSC is a cholangiopathy characterized by chronic fibroinflammatory damage of the large and extrahepatic bile ducts and is frequently associated with inflammatory bowel disease (IBD) [65]. Multiple simultaneous mechanisms appear to lead to PSC and its progression. One of these suggests that IBD may drive PSC rather than this being an epiphenomenon. Several lines of evidence propose a role for gut dysbiosis in the pathogenesis of PSC. The most accredited hypothesis proposes that a combination of environmental and genetic risk factors may induce biliary dysbiosis by rupturing the biliary mucosal barrier and generating toxic bile acid, which triggers bile duct fibrosis and cholangiocarcinogenesis. Dean et al. underline that abnormal enteric microbiome of PSC patients may promote the production of toxic products that are able to stimulate the immune-mediated damage of hepatocytes and the biliary tree by the migration of bacteria or associated toxins to the liver through the portal circulation [66]. Compared with IBD populations, PSC is characterized by a specific dysbiosis, but the difference between only PSC and PSC associated with IBD appears to be marginal, indicating that liver pathology is the principal corollary of microbial dysbiosis. In particular, in all of the studies we scrutinized, the genus *Veillonella* was enriched in the stool of PSC patients (Table 1). Other genera which frequently increase include: *Enterococcus*, *Streptococcus*, and *Lactobacillus*. Moreover, short-chain fatty acids producing anaerobes such as *Faecalibacterium* and *Coprococcus* were often found to be depleted in PSC patients [45]. Kummen et al., in a population of 136 patients with PSC (58% with IBD), 158 controls, and 93 patients with IBD without PSC, analyzed fecal DNA. The authors concluded that patients with PSC exhibited an increased prevalence of Clostridium species a depletion of Eubacterium spp and Ruminococcus obeum, an abundance of genes related to vitamin B6 synthesis and branched-chain amino acid synthesis, and reduced concentrations of vitamin B6 and branched-chain amino acids strongly associated with reduced liver transplantation-free survival [54]. Additionally, the salivary microbial signature of PSC is significantly altered, as demonstrated by Lapidot Y. et al., regardless of concomitant IBD, and includes 19 significantly altered species, of which eight species were consistently overrepresented in both fecal and saliva, as *Veillonella*, *Scardovia* and *Streptococcus*, and a significant overrepresentation of *Clostridium cluster XIVa* and *B. producta* in the gut microbiome could be observed [55] (Table 1). In experimental animal models, Nakamoto et al. identify a subset of microorganisms associated with PSC/UC as central mediators of bacterial translocation, immune regulation, and disease progression. In fact, their study showed that hepatic TH17 primed cells were observed only in PSC/UC mice, and that they exhibit an increased expression of hepatic inflammatory markers [67]. The authors also identified three species: *Klebsiella pneumoniae*, *Proteus mirabilis*, and *Enterococcus gallinarum*. The colonization of mice with *Klebsiella P*. might promote bacterial translocation via intestinal epithelial cell barrier dysfunction. However, its underlying mechanism remains unknown. To try to explain how the gut microbiota and the gut–liver axis are implicated in PSC pathogenesis, Liao et al. demonstrated in a Mdr2−/− mouse model that the loss of the phospholipid transporter Mdr2 triggers a cholestatic response, which induces intrahepatic sclerosing cholangitis. By analysis of Mdr2−/− microbiota, they demonstrated a reduced species diversity and significant alterations in the family of *Lachnospiraceae*, which have the important ability to form secondary bile acids and an increased activation of the NLRP3 inflammasome in dysbiotic Mdr2−/− mice with consequent disruption of intestinal barrier integrity and translocation of endotoxin into the portal vein [68]. 

A possible therapeutic approach for this pathology is represented by using antibiotics that are capable of removing harmful microorganisms and replenishing beneficial microbes or metabolites. Among the antibiotics studied, vancomycin, a glycopeptide antibiotic with Gram-positive activity, showed more efficacy data associated with the dihydroxylation of primary bile acids into secondary bile acids [69]. However, vancomycin is highly hydrophobic and toxic. Excessive concentrations of vancomycin have been linked to inflammation, cholestasis, and carcinogenesis. To determine whether antibiotic treatment modulates liver immune responses, Nakamoto et al., in their study previously cited, treated PSC/UC mice with either metronidazole or vancomycin that have antibacterial properties against K. pneumoniae and E. gallinarum, respectively. These PSC/UC mice showed a robust TH17 hepatic immune response that was significantly reduced with the use of either antibiotic. K. pneumonia should not be affected by vancomycin, so the decrease in TH17 in vancomycin-treated mice suggested the presence of additional pore-forming microbes targeted by vancomycin [67]. Tan LZ et al. describe their experience in treating 70 children affected by PSC and Ulcerative Colitis. In these patients the colitis of PSC–UC or autoimmune sclerosing cholangitis–ulcerative colitis (ASC–UC) can be severe and resistant to conventional therapy. Active colitis is a major risk factor for recurrent liver disease and graft failure. The authors concluded that oral vancomycin demonstrated excellent efficacy with regards to achieving clinical, biomarker, mucosal and histological remission of colitis in children with ASC and PSC [70]. Another recent report, published by Britto SL et al., underlined the abundance changes in Fusobacterium, Haemophilus, and Neisseria on longitudinal salivary and fecal microbiome changes in a pediatric PSC–UC patient over the first 90 days of vancomycin therapy [71]. In a systematic review and meta-analysis, Shah A. et al. evaluated the effect of antibiotic therapy in PSC patients. A total of 124 PSC patients received antibiotic therapy (57 treated with metronidazole, 35 with vancomycin, 16 each with rifaximin and minocycline). Treatment with antibiotics in PSC patients was associated with a statistically significant reduction in ALP, Mayo Risk Score, and total serum bilirubin level. Particularly ALP reduction was greatest for vancomycin and smallest with metronidazole, without significant adverse effects [72]. Other interesting antibiotics are rifaximin and minocycline. Rifaximin had no effect in decreasing cholestatic markers and the Mayo score in 16 PSC patients, as demonstrated by Tabibian et al. [73]. In contrast, minocycline was shown to cause an improvement In ALP levels and the Mayo score in 16 patients. Fecal Microbiome Transplantation (FMT) is a promising treatment for PSC patients. In one small pilot study, patients with PSC underwent FMT, and three of them experienced a ≥50% decrease in ALP levels. Its effect may be correlated with bacterial diversity and donor engraftment. The species that correlated with decreased ALP levels were Erysipelotrichaceae, Paraprevotella, Bacteroides and Alistipes taxa. Further studies are needed to establish the efficacy and safety of these therapies. 

Statin use is associated with improved outcomes of patients with PSC [74]. The rationale for this finding and for a current RCT with statins in PSC could be the fact that statin therapy is associated with a lower prevalence of gut microbiota dysbiosis [75]. 

As far as the biliary microbiome is concerned, Zigmond E et al. demonstrated, in a cohort of 189 patients affected by PSC undergoing endoscopic retrograde cholangiopancreatography (ERCP) for bile fluid collection, the presence of *Enterococci* that conferred risk of disease progression, such as fungobilia [58]. Additionally, Liwinsky et al. observed that the bile fluid of PSC patients was enriched by the presence of *Enterococcus faecalis*, a potentially pathogenic bacterium (Table 2) involved in an increase of the bile acid, taurolithocholic acid, which is known to be pro-inflammatory and potentially carcinogenic [60]. They also found that the disease duration is associated with an increased microbial burden, especially Fusobacterium and Gemella, which are associated with CCA development. Fusobacteria were increased during the long-term course of PSC. Miyabe et al. found that Fusobacterium nucleatum was statistically associated with PSC. However, in this study, statistical significance was not reached for IBD, obesity, or choledocholithiasis, according to their correlation analysis between the Fusobacterium level and these variables [76]. 

In a prospective non-randomized trial, Candida was detected in the bile of 7 out of 49 PSC patients with dominant stenosis. The authors also concluded that biliary Candida was associated with more severe cholangitis [59] (Table 2).

## 7. CCA

In recent years, the term “oncobiome” has been introduced to indicate the implication of gut microbiota in the development of neoplastic diseases [77]. To date, the most probable hypotheses through which the microbiome influences cancer development are related to the modulation of host local and systemic immune responses by the microbiota [78], the role of bacterial toxins on carcinogenesis [79], and modifications in microbial and host metabolism [80]. 

CCA is a malignant tumor that originates from the epithelial cells of the PBGs [81]. It accounts for 3% of gastrointestinal cancers, although it is considered a rare tumor [82]. The incidence is lowest in Eastern countries, where the number of cases is less than 5/100,000 people/year. [83] However, the burden of CCA is underestimated worldwide as it is the leading cause of metastasis of unknown origin [84]. Surgery is the only potentially curative strategy for CCA. In advanced forms, 5-year survival is lower than 15% [83]. From an anatomic point of view, CCA can be classified into intrahepatic (iCCA), peribiliary (pCCA), and distal (dCCA). This classification reflects a histological differentiation as pCCAs and dCCAs are more frequently mucinous adenocarcinomas, unlike iCCAs [81]. Different epidemiological studies highlight that the several risk factors implicated in cholangiocarcinogenesis are different between iCCA and the others [85]. Established risk factors for CCA are parasitic infections, biliary tract diseases (such as primary sclerosing cholangitis (PSC) and Caroli disease), and hepatolithiasis. Other risk factors are liver cirrhosis, hepatitis B, hepatitis C, diabetes mellitus, obesity, alcohol, and smoking [83]. CCA may develop through the accumulation of epigenetic modifications that occur due to exposure to environmental and microbial factors. The biliary ducts are vulnerable to the gut microbiome by way of the gut–liver axis [15] and, as for other tumors, on top of the already known risk factors, recent experimental studies have shown that intestinal microbial dysbiosis may also be involved in the development of CCA [62,86,87] (Table 2). This is probably related to the pathway of bile acid metabolism. 

Recently, some studies have highlighted that CCA has a different gut microbiome, even compared to the other primary liver tumor, the hepatocellular carcinoma. The study conducted by Deng et al. showed that patients with CCA had a higher α-diversity than hepatocellular carcinoma (HCC) patients, while HCC patients had decreased α-diversity in comparison to controls. This presented the opportunity to use the gut microbiome as a diagnostic instrument in liver cancer [56]. In another study, Zhang et al. studied the discrepancy in gut microbiota between CCA patients, patients with cholelithiasis, and healthy patients (Table 1). CCA patients and healthy patients had a more species-rich and homogeneous microbiota than the cholelithiasis group, while there were differences in α-diversity between healthy controls and CCA patients [57]. Jia et al. have recently shown that the gut microbiome is closely related to the tumorigenesis and progression of CCA; furthermore, the intestinal microbiome of the patients affected by iCCA had the highest diversity [88] (Table 1). 

In comparison with other categories, at the genus level, Alloscardovia, Peptostreptococcaceae, Actinomyces, and Lactobacillus were remarkably increased in the patients affected by iCC. Furthermore, differences in the tissue microbiome were found in liver fluke-associated CCA and non-liver fluke-associated CCA [89]. In this regard, it is known that *Opisthorchis Viverrini* infection is a risk factor for CCA. Studies highlighted that this infection is responsible for modifying the gut microbiota. In particular, in these patients, it has been marked with an increase of *Helicobacter* spp. in stool samples and an overexpression of two genes, Cag A and Cag, which are implicated in promoting inflammation and fibrosis of the bile ducts [90].

Regarding the various species of Helicobacter pylori, the most significant studies include two meta-analyses and three prospective studies. The two meta-analyses examine ten case–control studies. Both of these analyses expose a significant relationship between Helicobacter species and the presence of CCA compared with control patients with benign diseases of the biliary tract or without pathologies [63,91]. Furthermore, these analyses highlighted an increased rate of H. pylori (49.5% vs. 33.3%, *p* = 0.003) and H. bilis (52.2% vs. 23.7%, *p* < 0.0001) in patients affected by CCA compared to those affected by benign biliary diseases [62]. The three prospective studies confirm the association between Helicobacter species and CCA, as already shown by the meta-analyses [62,64,92]. 

Other studies investigated the possible association between CCA and microbiota. There are fourteen significant studies [93]. Different sample types from patients with BTC were analyzed: bile, fecal, plasma, and tissue samples. Microbiota was analyzed through an automated microbiology system (Phoenix or Vitek-2), 16S rRNA gene sequencing or shotgun metagenomics. The stage of CCA was not mentioned in most of the studies. These studies report an increase in Fusobacteria (four studies), Enterobacteriaceae (three studies), and Pseudomonadaceae (three studies) in patients with biliary tract cancer (CCA and gallbladder carcinoma), HCC, and pancreatic head tumors [93].

At the same time, a great amount of evidence hints that the bile microbiota plays a fundamental role in diseases of the biliary tract, including CCA (Table 2). Avilés reported that, for the first time, the microbiota in the biliary tract in cancer and non-cancer conditions found a significant increase in the genera Novosphingobium, Actinomyces, Fusobacterium and Prevotella, and a decrease in Nesterenkonia and Rothia in ECCA [56]. In a recent study, Saab et al., have described the biliary microbiota in extrahepatic CCA (eCCA) patients and compared them with controls. They identified a dysbiosis that was significantly related to eCCA. This included genera such as Bacteroides, Geobacillus, Anoxybacillus, and Meiothermus, which were found to be more abundant in cases than in controls [61]. Even Miyabe et al. have recently found significant differences in the bile microbiome in PSC and CCA. Specifically, it was observed that long-lasting PSC increase the amount of bile microbioma associated with an increase in specific bile microbioma characteristics. This may increase inflammation in the biliary tract and influence the risk of CCA in patients with PSC [76]. Aside from those previously mentioned, several other studies have investigated the association between CCA and specific micro-organisms [76].

## 8. Conclusions

In recent years, in addition to the concept of gut microbiota referring to pathogenic bacteria and other microorganisms that colonize the gastrointestinal tract, several discoveries have been made regarding the biliary microbiome and the close communication between these two environments, their condition of balance through the enterohepatic circulation, and the state of immune tolerance. Under conditions of altered equilibrium, bacterial products deriving from the intestine induce the activation of strong immune responses in cholangiocytes which may contribute to the development of biliary lesions. At the same time, other studies are gradually delineating a specific composition of the gut and biliary microbiota in patients with cholangiopathies that could be partly responsible for the activation of cholangiocytes. Advances in the knowledge and understanding of the gut–liver axis and biliary–liver axis are of fundamental importance because they are driving the development of diagnostic, prognostic, and therapeutic tools based on microbiota to manage cholangiopathies, such as FXR agonists and antibiotics.

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
