# Peer review of "Role of the Gut–Liver Axis in the Pathobiology of Cholangiopathies: Basic and Clinical Evidence"

_ijms, 2023, doi:10.3390/ijms24076660_

Round 1

Reviewer 1 Report

This is a nice review of the relationship between the gut and the liver in the development of cholangiopathies, which covers the main points well

There is however, many minor errors in the English. Some examples are listed below, but many more occur throughout the manuscript. 

Line 31-34 – should be a comma, not a semi-colon. The word they is unnecessary. Should say “generating a physical and chemical barrier to protect the mucosa from commensal and pathogenic microorganisms

Line 34 – (3) should be before the full-stop

Line 42 – should pathogen be pathogenic?

Line 44 – should microbioma be microbiome?

Line 62 – increase the permeability of the paracellular pathway?

Line 78, should be pathogens

Lines 85-87. This sentence doesn’t make sense, and needs referencing

Line 87 -  should say recent evidence not evidences

 Line 197 - should say As far as the gut microbiota is concerned 

Author Response

Dear reviewers,

as you indicated we have taken steps to approvate the requested changes using the "Track change" function so that they can be more easily individualized. 

You will also notice, as written in a comment in the text, that as you recommended a word has not been modified as the entire sentence has been modified

Thank you

Kind regards

Maria Consiglia Bragazzi

Reviewer 2 Report

This manuscript by Bragazzi et al examines the function of the gut-liver axis in the pathogenesis of cholangiopathies. The Gut-Liver Axis is the interaction between the gut, its microbiota, and the liver in both directions. Immune tolerance maintains this interaction in a healthy condition. Nevertheless, research demonstrates that alterations in gut bacterial homeostasis or impairment of intestinal barrier functions can cause cholangiocytes (epithelial cells lining bile ducts) to activate innate immune responses against gut-derived microorganisms or bacterial products that reach the liver via enterohepatic circulation. This exposure to a growing number of microorganisms might result in inflammatory responses that aggravate liver fibrosis. 

Comments-

1- A figure describing the role of gut microbiota and its metabolites in cholangiopathies needs to be added to this review.  

2- In addition to the PBS, PSC, and CCA the authors also need to discuss the other types of cholangiopathies including polycystic liver diseases (PLD) and Biliary atresia.

Author Response

Dear reviewer,

attached is a figure describing the bile barrier and defensive system

Kind regards

Maria Consiglia Bragazzi

Round 2

Reviewer 2 Report

The author's response to the reviewer's comment letter is not attached to this manuscript. I see a figure instead of a response letter. Manuscripts with changes made by authors also need to be submitted.

Author Response

Dear reviewer, 

to date there is no data in the literature on the role played by the gut microbiota in the other types of cholangiopathies  

Best regard

Maria Consiglia Bragazzi

Round 3

Reviewer 2 Report

I am now satisfied with this manuscript.